



# Improved simulation of clouds over the Southern Ocean in a General Circulation Model

Vidya Varma[1], Olaf Morgenstern[1], Paul Field[2], Kalli Furtado[2], Jonny Williams[1], and Patrick Hyder[2]

[1]National Institute of Water and Atmospheric Research (NIWA), Wellington, New Zealand
[2]Met Office, Exeter, United Kingdom

**Correspondence:** Vidya Varma (vidya.varma@niwa.co.nz)

**Abstract.** The present generation of global climate models is characterized by insufficient reflection of short-wave radiation over the Southern Ocean due to a misrepresentation of clouds. This is a significant concern as it leads to excessive heating of the ocean surface, sea surface temperature biases, and subsequent problems with atmospheric dynamics. In this study we modify cloud micro-physics in a recent version of the Met Office's Unified Model and show that choosing a more realistic value for

the shape parameter of atmospheric ice-crystals, in better agreement with theory and observations, benefits the simulation of short-wave radiation. In the model, for calculating the growth rate of ice crystals through deposition, the default assumption is that all ice particles are spherical in shape. We modify this assumption to effectively allow for oblique shapes or aggregates of ice crystals. Along with modified ice nucleation temperatures, we achieve a reduction in the annual-mean short-wave cloud radiative effect over the Southern Ocean by up to $\sim 4\,\mathrm{Wm}^{-2}$, and seasonally much larger reductions. By slowing the growth of

the ice phase, the model simulates substantially more supercooled liquid cloud. We hypothesize that such abundant supercooled liquid cloud is the result of a paucity of ice nucleating particles in this part of the atmosphere.

## 1   Introduction

One of the major known drawbacks in the present-day global climate models is an excess in the absorbed short-wave (SW) radiation over the Southern Ocean (SO) (Trenberth and Fasullo, 2010; Ceppi et al., 2012; Hwang and Frierson, 2013; Hyder

et al., 2018). In Chapter 9 of the 5[th] Assessment Report of the Intergovernmental Panel on Climate Change (IPCC AR5) (Flato et al., 2014), it points out that most of the 5[th] Coupled Model Intercomparison Project (CMIP5) models (Taylor et al., 2012) have a positive SW cloud radiative bias of magnitude of up to $20\,\mathrm{Wm}^{-2}$ over the SO, suggesting that inadequately simulated clouds allow substantially too much sunlight to reach the ocean surface.

Several studies have focused on the relation between various aspects of cloud representation in the model and observed

radiation biases. Bodas-Salcedo et al. (2012, 2014), using cyclone compositing cluster analyses, suggest the need to increase the optical depth of the low-level clouds and improve the simulation of mid-level cloud regime, to help reduce the biases in the model. By modifying the shallow convection detrainment in their global climate model, Kay et al. (2016) showed that the resultant increase in the supercooled liquid clouds (SLC) enable large reductions in long-standing climate model SW radiation biases. By implementing a new parametrisation that includes the turbulent production of mixed-phase clouds, Furtado et al.





(2016) show that the radiation biases can be substantially improved, especially over the SO. In another study by Furtado and Field (2017), the importance of ice micro-physics parametrisation in determining the phase composition, and thus the liquid water content of the SO clouds is highlighted.

Discrepancies in the response of clouds to anthropogenic forcings are recognized as a leading reason for a persistent, large

spread in the climate sensitivity throughout various generations of climate models (Allen et al., 2014). We thus conjecture that this model problem contributes to this large spread, and thus solving it would increase confidence in projections of anthropogenic climate change (Tan et al., 2016).

In the present study, we investigate the role of parameters involved in atmospheric ice formation within a global climate model in causing the above mentioned SW radiation bias. Here, we define a SO region as the latitudinal band between 50°S

and 70°S.

## 2 Data and experimental set-up

The control climate model used in this study is an accrual of the most recent version of the Met Office's Unified Model, GA7.1 (Walters et al., 2019) with modified micro-physics scheme for riming process and several other scientific changes. Appendix A summarizes the scientific set-up for this model version. The resolution used here is N96L85 (i.e. a horizontal resolution of

$1.875° × 1.25°$ and 85 terrain-following hybrid-height levels extending to 85 km of altitude). It uses the "ENDGAME" dynamical core with a semi-implicit semi-Lagrangian formulation to solve the non-hydrostatic, fully compressible deep-atmosphere equations of motion (Wood et al., 2014)

### 2.1 Model set-up

In the present study, control run follows the Atmospheric Model Intercomparison Project (AMIP) climate model development

protocol (Gates et al., 1999; Schuddeboom et al., 2019), using prescribed sea-surface temperature climatology. Excess atmospheric ice has been a persistent concern in the control version of the model (fig. 1), which is especially pronounced over the SO region. Ice clouds have a significant influence on the global climate through their effects on the Earth's radiation budget e.g. (Hartmann and Doelling, 1991; Waliser et al., 2009). Hence, sensitivity set-ups in our study are aimed at modifications to the micro-physics scheme such that the ice growth in the model is controlled. We achieve this by modifying those parameters

that control the growth of existing ice by vapor deposition and heterogeneous nucleation of new ice. The classical theory of ice crystal growth uses an electrostatic analogy due to the similarity between the equations governing the water vapor distribution around an ice crystal and the electrostatic potential distribution around an electric conductor of the same shape as the ice crystal (Chiruta and Wang, 2003). Thus, the growth rate of ice crystals by diffusion depends on a shape (also known as capacitance) parameter $C$, which is a function of both ice crystal size and habit. To determine the ice crystal growth rates in models, it is

necessary to know the value of $C$ (Chiruta and Wang, 2003; Hobbs, 1976). The standard equation that is used for calculating the growth rate of ice crystals in the model is,

$$\frac{dm}{dt} = 4\pi D_v C (\rho_\alpha - \rho_s) \tag{1}$$





where $D_v$ is the diffusivity of water vapor in air, $C$ is the capacitance, $\rho_\alpha$ and $\rho_s$ are distributions of vapor densities at and away from crystal's surface.

From eq.1, it is evident that once the value of capacitance $C$ is known, the growth rate of ice crystals can be determined. All other quantities on the right-hand side of eq.1 are independent of the shape (Chiruta and Wang, 2003; Hobbs, 1976). Thus, $C$ in

the model effectively defines the shape of ice crystals, which in turn is fed through to the ice processes of deposition/sublimation and melting without affecting any other ice processes.

Technically the capacitance, $C$, is defined as 1.0 x $d$ in the model, where $d$ is the particle maximum size. In our sensitivity studies, we modified the value to 0.5 x $d$ (corresponding to any oblate ellipsoid with two unequal axes, thought to be more appropriate for aggregates and plate-like crystals rather than the assumption of spherical crystals alone). Our value of 1.0 or 0.5

is a non-dimensional capacitance (Field et al., 2008). The effect of this change in the shape parameter is tested independently as well as in combination with changing the temperatures at which heterogeneous and homogeneous freezing start in the cloud micro-physics scheme. The ice nucleation temperature is the temperature at which heterogeneous nucleation of ice first starts to occur in the model. The default value of $-10°$C was changed to $-40°$C and $-20°$C, to investigate the effect of delaying the heterogeneous ice nucleation in the model. Two further parameters that were modified in the parametrised convection scheme

that control ice formation in the model are the temperature at which detraining condensate as ice begins in the model (start-ice temperature) and the temperature at which all condensate is detrained as ice (all-ice temperature). The values used in our numerical simulations are summarised in Table 1.

The ice nucleation temperature is meant to be similar both in large-scale and convective cloud schemes. Hence, the experiment where the nucleation temperature is reduced to -40°C (i.e. exp2) is physically unrealistic. However, it is still a useful

sensitivity scenario to study the importance of detrained ice vs. large-scale freezing. All simulations were run for twenty years under steady-state present-day conditions.

## 2.2 Observational data

We use the National Aeronautics and Space Administration (NASA) Clouds and the Earth's Radiant Energy System - Energy Balanced And Filled (CERES EBAF Ed4.0, Terra-Aqua) surface and top-of-the-atmosphere (TOA) data set, covering the period

2000 to 2018 as an observational reference for radiative fluxes. This data set, in an earlier version (Loeb et al., 2009), was also used in AR5. The overall uncertainty in the monthly all-sky TOA flux for the CERES EBAF Ed4.0 data set is estimated to be 2.5 Wm$^{-2}$ (for for both SW and LW fluxes). For clear-sky TOA, uncertainties in SW and LW fluxes are 5 Wm$^{-2}$ and 4.5 Wm$^{-2}$ respectively (Loeb et al., 2018). We also use the European Centre for Medium-Range Weather Forecasts (ECMWF) Re-Analysis 5 (ERA5) monthly mean data for comparison of cloud-ice content (ERA5, 2017).

## 3 Results

Fig.2 represents the anomaly in the annual and DJF mean distributions of ice water path (IWP) and liquid water path (LWP) for stratocumulus boundary layer clouds in the model in various experiments with respect to the control run, for the Southern





Hemisphere (SH). The boundary layer types have been identified based on the surface stability and capping cloud (Lock et al., 2000). Further information on the types of boundary layers considered is included in the figure caption.

From fig. (2a), it is evident that there is noticeable decrease in the annual-mean IWP in the boundary layer clouds as a result of modified micro-physics, which is captured in all sensitivity experiments. Exp2 (solid black line in fig. 2a) shows
the maximum response and exp1 (solid red line fig. 2a) has the minimum decrease in IWP with respect to control run . This response pattern is similar for DJF mean as well (fig. 2b). Conforming to the decrease in the IWP, there is a corresponding increase in the LWP as well, over the SO region (figs. 2c and 2d). However, the response of LWP is more or less similar in all the 3 experiments w.r.t the control run. It is because modification to capacitance value affects both the liquid and ice water contents while changes to nucleation temperature will have an impact predominantly on ice water path. Thus, experiments
where both capacitance and nucleation temperature are modified will have an added impact on the IWP.

Zonally averaged distribution of IWP and LWP, over both hemispheres, for all boundary layer types for annual and seasonal means are provided in the supplementary material (figs. S1 to S4).

Figure 3 shows the zonal-mean changes in the annual-mean distributions of various radiative fluxes in the model for the SH. In all the model experiments there is a general decrease in the outgoing long-wave (LW) flux at the top-of-the-atmosphere
(TOA) in the SO region (solid red line in figs. 3a to 3c) with respect to the control run. This is accompanied by a corresponding increase in the outgoing SW flux at the TOA (solid black line in figs. 3a to 3c), indicating that in all experiments the planetary albedo has increased versus the control. Except in exp1 (i.e. fig. 3a), the decrease in LW radiation at the TOA, in absolute terms, is larger than the increase in SW TOA over the SO. This is visible in the distribution of net radiation at the TOA (i.e. LW plus SW at TOA) as well (solid mustard lines in figs. 3a to 3c). For exp1, there is an increase in the net outgoing TOA
radiation whereas for exp2 and exp3, it shows a decrease over the SO region.

The surface distributions of the radiative fluxes are represented by solid magenta, gray and blue lines in figs. 3a to 3c. In all the experiments, the net downward LW radiation at the surface shows an increase over the SO (solid magenta lines). The corresponding SW component shows a decrease over SO (solid gray lines). The net radiation at surface (i.e. downward LW plus SW at surface) shows a general decrease in all experiments over SO (solid blue lines). The distribution of anomaly in the
25 radiative heat flux with respect to the control run is represented by solid cyan line in figs. 3a to 3c. It primarily represents the difference between total net downward surface radiation and total heat flux at the surface i.e (LW + SW) - (sensible heat flux, SH + latent heat flux, LH). Although there is an improvement (i.e. reduction) in the downward SW component (solid gray lines), due to the compensating increase in the LW component (solid magenta), there is a net increase in the heat flux into the surface over the SO (solid cyan lines) in almost all experiments. However, the net radiative heat flux shows a slight decrease
over SO for exp1 w.r.t the control run (solid cyan line in fig. 3a).

Figure 4 shows the distributions of various radiative fluxes in the model for the SH for the DJF season. As expected, the radiative fluxes show a more pronounced response during the austral summer season. The net radiative flux at TOA (solid mustard lines) is showing an increase over the SO for all the experiments unlike the annual-mean distribution where only the exp1 showed an increase. Similarly, the net radiative heat flux at the surface (solid cyan lines) shows a general decrease over
35 the SO in all the experiments w.r.t the control run for the DJF season whereas in annual-mean, only exp1 showed a decrease.





The dashed lines in fig. 3d represent the difference between the observational data and model control data for annual-mean. It is mainly intended to provide a reference for the model behaviour in terms of radiative fluxes. The surface radiative fluxes in the model (dashed magenta, gray, blue and cyan lines) are generally in better agreement with the observations than those at the TOA (dashed red, black and mustard lines), especially over the SO region. Although the SW TOA flux (dashed black

line) is mostly in agreement between model and observational data, the disparities in LW TOA equivalent (dashed red line) are noticeable in the net radiative flux at surface well (dashed mustard). The dashed lines in fig. 4d, represent the observational data reference for the DJF season. Relative to the annual-mean, the model and observational data are more comparable in terms of the signs of response in the DJF season.

Global zonal plot for annual and seasonal mean distribution of radiative fluxes are provided in the Supplementary material

(figs. S5 to S7). Supplementary material also includes the anomaly of all model experiments w.r.t the observational data as well, i.e. model experiments - observational data (figs. S8 to S10).

Figure 5 shows the zonal averaged distribution of annual and DJF mean distributions of the anomaly in the SW cloud radiative effect (SW CRE) between different model runs with respect to the control run. The SW CRE is calculated by differencing the upwelling SW radiation in cloudy and non-cloudy conditions (Ramanathan et al., 1989; Allen et al., 2014). It is evident

that there is significant improvement (i.e. a reduction) in the SW CRE over the SO regions in all three experiments compared to the control run. The reduction in the SW radiative flux over SO is more pronounced in the DJF season (fig. 5b). For both annual and DJF means, both exp1 and exp3 (solid red and yellow lines in 5a and b) show a stronger reduction in SW CRE over the SO region than exp2 (solid black line) compared to the control run.

The zonal SW CRE for austral winter season is provided in the supplementary material (fig. S11).

Figure 6 shows the evolution of SW CRE improvement over SO in various versions of the UM. Figs. 6a and 6b show the anomaly in the SW CRE in the previous model versions of GA6 (Walters et al., 2017) and GA7 (Walters et al., 2019) w.r.t that of the CERES EBAF observational data. As evident, there has been significant improvements of the SO radiation biases in GA7 compared to its predecessor GA6. One of the major reasons behind this improvement was a better representation of mixed-phase clouds and supercooled liquid in the cloud micro-physics scheme (Furtado et al., 2016). Fig. 6c shows the SW

CRE comparison of the current control model version, with that of the observational data. In general there is an increase in the reflected SW radiation in the current control model version compared to GA7. While this has benefited regions like equatorial precincts of Indian, western Pacific, North Atlantic and obviously the SO, it also has some adverse impact on regions like equatorial eastern Pacific, South Atlantic etc. However, sectors like the Bay of Bengal, South China Sea etc still have biases just like in the previous model versions.

Figures 7a to 7c show the SW CRE anomaly in various experiments used in this study w.r.t that of the control run. As expected, all the experiments show a decrease in the SW CRE compared to the control run over the SO region. As already shown in fig. 5a, the reduction of SW CRE over SO is most pronounced for exp1 and exp3 followed by exp2. For exp1 (capacitance only), there is an increase in the SW CRE over regions like eastern Australia, South China Sea, eastern sects of South America etc. For exp2 and exp3, equatorial western Pacific shows much sensitivity in terms a reduced SW CRE

compared to the control run.



## 4 Discussion

An overestimation of ice in clouds is a known shortcoming of many of the present-day global climate models (fig. 1). It is coupled to an underestimation of SLC. This problem is of particular importance in the SO region characterized by abundant SLCs (Kay et al., 2016; Bodas-Salcedo et al., 2016; Huang et al., 2012; Hu et al., 2010). When ice and supercooled liquid

coexist, the ice grows at the expense of the liquid by the Wegener-Bergeron-Findeisen (WBF) mechanism (Wegener, 1911; Bergeron, 1935; Findeisen, 1938). Acknowledging the complexities in representing the many possible background microphysical processes that are responsible for this in a global climate model, the primary idea of modifying the shape parameter of ice-crystals is to reduce the rate of depositional growth of ice particles. This reduction essentially slows down the deposition growth of ice crystals, which leaves more water vapor to be available for condensation into liquid phase particles. At the scale

represented in global climate models, for conditions of very low ice-nucleating particle (INP) concentrations and temperatures between the homogeneous and the heterogeneous freezing points, this essentially amounts to a limitation of the speed of glaciation as freezing of SLCs can only occur at the interfaces of these two states. By lowering the value of capacitance to 0.5, we model a decrease in the IWP and an associated increase in the LWP over the SO region (fig. 2). As a result of this increase in LWP, the outgoing SW fluxes are increased (solid black lines in figs. 3 and 4), i.e. an increased LWP corresponds to brighter

clouds reflecting more sunlight. This results in a decrease of the downwelling short-wave radiation reaching the surface (solid gray lines in figs. 3 and 4).

Our choice of $0.5 * d$ (d being the particle maximum size) for capacitance is based on theory and observational studies (Field et al., 2008). The atmosphere-only model studied here does perform better with this value than with the default value of $1.0 * d$. The SW radiation over SO is improved but results are more mixed for the other fluxes (figs. 3 and 4). The uncertainty

in the surface radiation budget observations also needs to be considered. As already noted earlier, the experiment where the nucleation temperature is reduced to -40°C (i.e exp2) is physically unrealistic and is intended to be a useful sensitivity scenario to study the importance of detrained ice vs. large-scale freezing.

Even though there is noticeable reduction in the SW radiation bias over the SO in all the experimental scenarios (fig. 5), we recognize persisting shortcomings in this regard in other parts of the world (figs. 6c. Similar to the control version in this study,

certain regions have not shown much of an improvement in terms of the SW CRE bias (e.g. the Bay of Bengal, areas around southeast Asia, eastern south Pacific etc). Previous studies have suggested that some cloud micro-physics parameterisations produce unrealistically bright clouds, especially over the Northern Hemisphere (NH) (Furtado et al., 2016). Since the SW biases over the NH were smaller than those over the SO, a significant brightening of modelled NH clouds is undesirable (Furtado et al., 2016). While the changes to nucleation temperature has significant impact on the tropics as well, the capacitance changes

are more localized to the high latitudes.

Several recent studies point towards the significance of INP for cloud phase (Kanji et al., 2017; Vergara-Temprado et al., 2018). A further development of the research outlined here could be to make glaciation explicitly dependent on INP concentration. At present, in most global climate models, cloud phase is determined only by a threshold temperature. Vergara-Temprado et al. (2018), using a high-resolution numerical weather prediction model and making assumptions on the INP concentration





over the SO, simulate clouds that are for more reflective than those in current global climate models, in better agreement with satellite observations.

# 5    Conclusions

In this study we improve the SW radiation biases in a recent version of the UK Met Office's Unified Model. This and other
contemporary climate models are characterized by excess cloud ice causing biases in SW radiation biases which are especially pronounced over the SO. Here, we modify the capacitance or shape parameter which represents ice crystal shape and habit. In our sensitivity studies, we reduce this parameter from 1 to 0.5 (corresponding to any oblate sphere shape in general, where the horizontal axes are longer than the vertical axis and more representative of an aggregate or flat ice crystal). We also examine the impact of changing other temperature thresholds in the cloud micro-physics scheme for the onset of heterogeneous ice
production. Our analysis shows that the SW radiation bias has significantly reduced over the SO after the modification of these parameters. However, disparities still exist in other regions. INPs that are currently not represented in the cloud micro-physics scheme might be a factor in this model behavior. The fact that nucleation temperature changes currently is associated with the same effects globally is undesirable, it further motivates the future work to couple the nucleation temperature to a prognostic or the least a regionally specified INP concentration.

*Data availability.* Model data is available at:http://doi.org/10.5281/zenodo.3229446.

Observational data is available at: https://ceres.larc.nasa.gov/order_data.php and

https://cds.climate.copernicus.eu/cdsapp#!/dataset/reanalysis-era5-single-levels-monthly-means?tab=form

# Appendix A

Several changes were introduced in the GA7.1* control model used in this study relative to its predecessor GA7.1 (Walters
et al., 2019; Brown et al., 2012). These changes range from minor bug fixes and optimisation techniques to major science changes. As far as our study is concerned, the main modification to GA7.1 is the inclusion of the modified micro-physics scheme which includes a shape dependence of riming rates using the parameterization by Heymsfield and Miloshevich (2003), as a measure to prevent small liquid droplets from riming (Furtado and Field, 2017). The reference link to the control model with all scientific/technical details is documented in a Met Office internal repository ticket: GA7.1#256
https://code.metoffice.gov.uk/trac/gmed/ticket/256. A brief description of scientific changes between various model versions can also be found in (Bodas-Salcedo et al., 2019).

*Author contributions.* VV carried out the model runs, performed analysis, created figures, wrote the manuscript and was also involved in the design and conceptualization of the study. OM was involved with obtaining the project grant, supervised the study and analyses of results.





PF, KF and PH provided guidance in designing the model runs and analyses of results. JW provided technical support in setting up global climate model in super-computer environment. All authors have read and approved the final paper.

*Competing interests.* The authors declare that they have no conflict of interest.

*Acknowledgements.* This work has been funded by the New Zealand Government Ministry for Business, Innovation, and Employment
5 (MBIE) through the Deep South National Science Challenge. This work has also been supported by NIWA as part of its government-funded core research. The authors wish to acknowledge the use of New Zealand eScience Infrastructure (NeSI) high performance computing facilities, consulting support and training services as part of this research. New Zealand's national facilities are provided by NeSI and funded jointly by NeSI's collaborator institutions and through the Ministry of Business, Innovation & Employment's Research Infrastructure programme. https://www.nesi.org.nz.





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





**Table 1.** Values used in the model runs

|  | **Capacitance** | **Ice nucleation temperature (°C)** | **Start-ice temperature (°C)** | **All-ice temperature (°C)** |
|---|---|---|---|---|
| **control** | 1.0 | −10 | −10 | −20 |
| **exp1** | 0.5 | −10 | −10 | −20 |
| **exp2** | 0.5 | −40 | −40 | −41 |
| **exp3** | 0.5 | −20 | −40 | −41 |

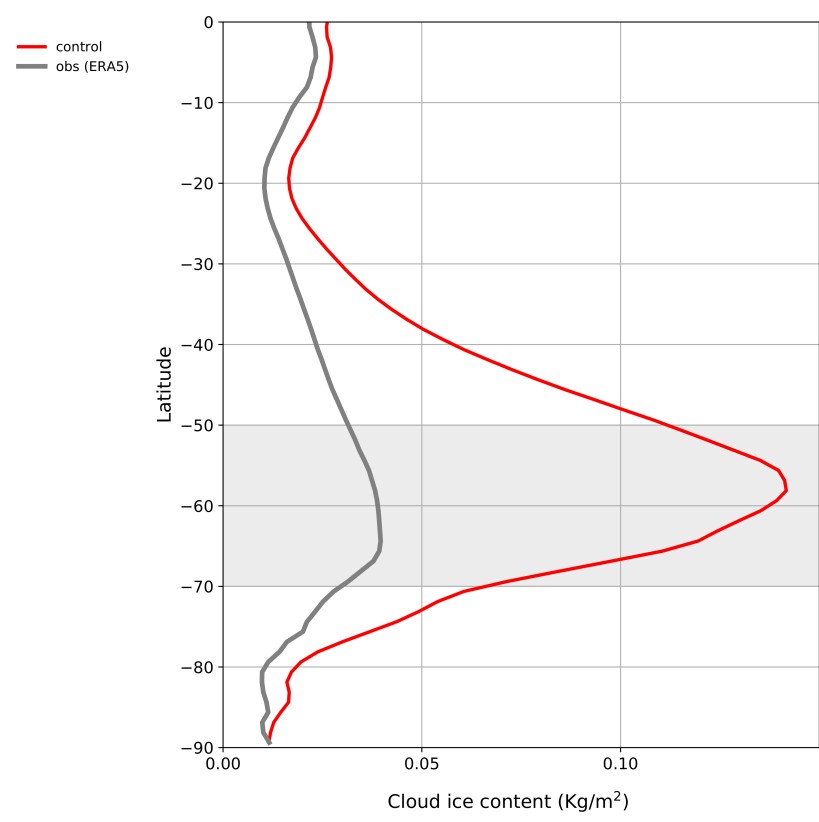

**Figure 1.** Annual-mean zonally averaged cloud ice content in the control model (solid red line) and ERA5 observational data (solid gray line) for the Southern Hemisphere. The SO region identified in this study is highlighted in gray.

**Figure 2.** Distribution of zonally averaged anomalies in IWP (solid lines in 2a and 2b) and LWP (dashed lines in 2c and 2d) over the stratocumulus boundary layer type clouds in the model for the SH. The cloud types considered in the model are: type 2 = boundary layer with stratocumulus over a stable near-surface layer, type 3 = well-mixed boundary layer and type 4 = unstable boundary layer with a decoupled stratocumulus (DSC) layer not over cumulus. The IWP and LWP are calculated collectively over these types. (2a) and (2c) represent annual-mean; (2b) and (2d) represent DJF mean. The colour codes are as follows: red = anomaly of exp1 with respect to control, black = anomaly of exp2 with respect to control, yellow = anomaly of exp3 with respect to control. Values are calculated from 12 hourly model output over 20 years. The SO region identified in this study is highlighted in gray.

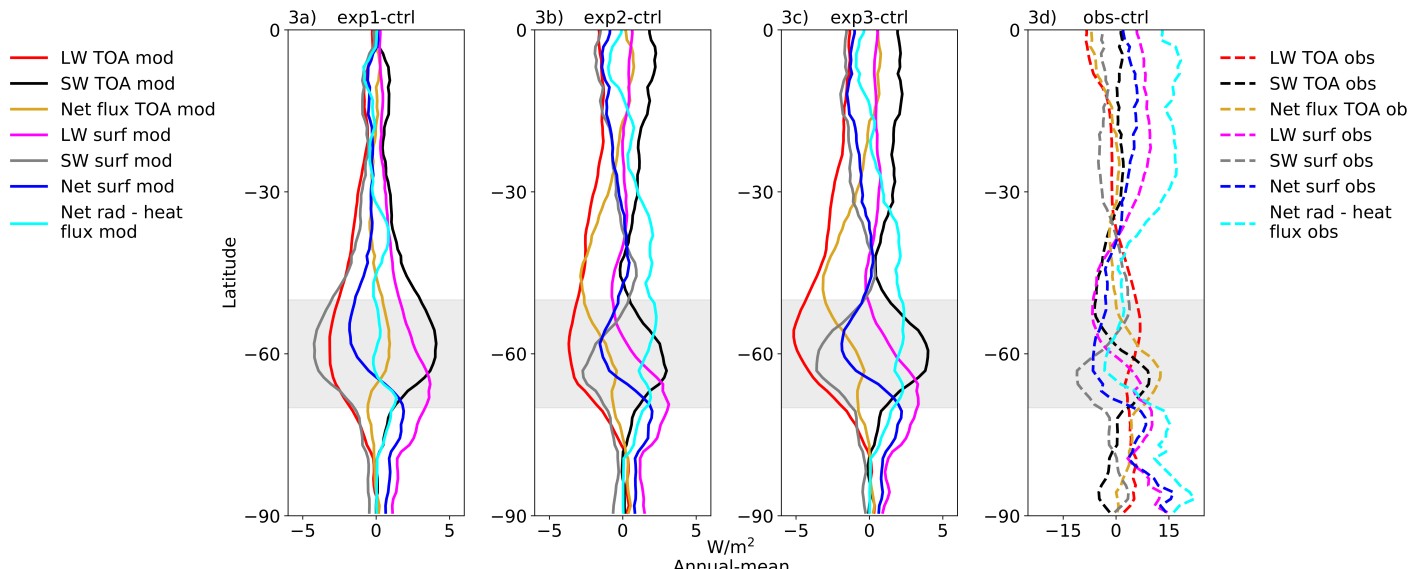

**Figure 3.** Distribution of zonally averaged annual-mean radiative flux anomalies in various experiments and observational data with respect to the model control run for the SH. 3a) to 3c) LW and SW radiative fluxes (leaving the TOA as well as the net downward at the surface) for exp1 - control, exp2 - control and exp3 - control respectively. 3d) similar to (3a) but for observational data - control. The colour codes are as follows: red = LW TOA, black = SW TOA, mustard = LW TOA + SW TOA, magenta = LW surface, gray = SW surface, blue = LW surface + SW surface, cyan = (LW surface + SW surface) - (sensible heat + latent heat). Solid lines represent radiative flux anomalies from model and dashed lines represent anomaly of observational data w.r.t control run. Annual-mean values for model are calculated from 12 hourly output over 20 years. Observational data consist of monthly mean values covering the period 2000-2018. The SO region identified in this study is highlighted in gray.





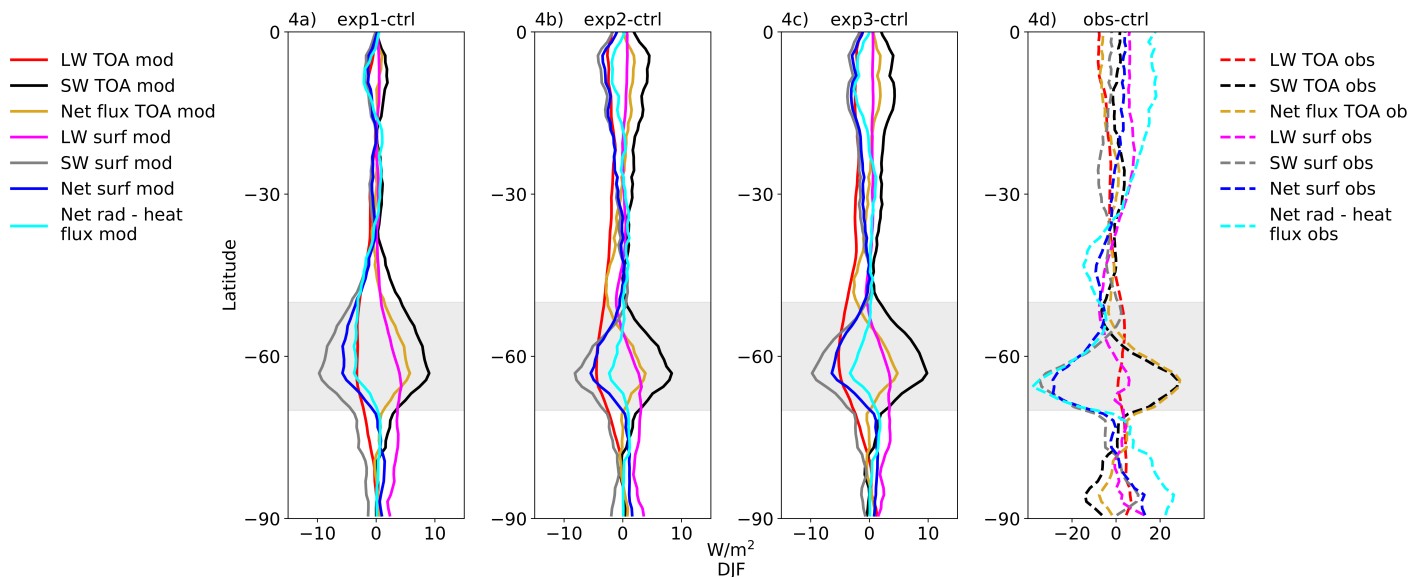

**Figure 4.** Similar to Figure 3 but for DJF season.



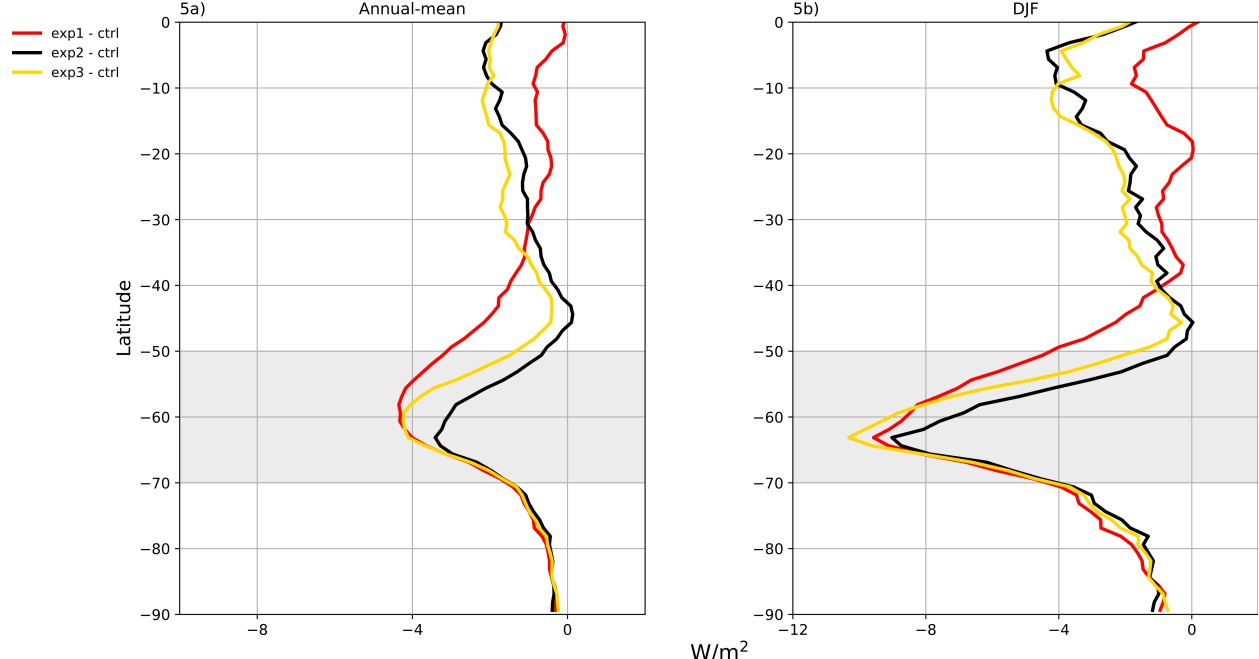

**Figure 5.** Distribution of zonally averaged SW CRE anomalies over SH in various experiments w.r.t the model control run for (5a) annual-mean and (5b) DJF mean. The colour codes are as follows: red = anomaly of exp1 with respect to control, black = anomaly of exp2 with respect to control, yellow = anomaly of exp3 with respect to control. Values are calculated from 12 hourly output over 20 years. The SO region identified in this study is highlighted in gray.





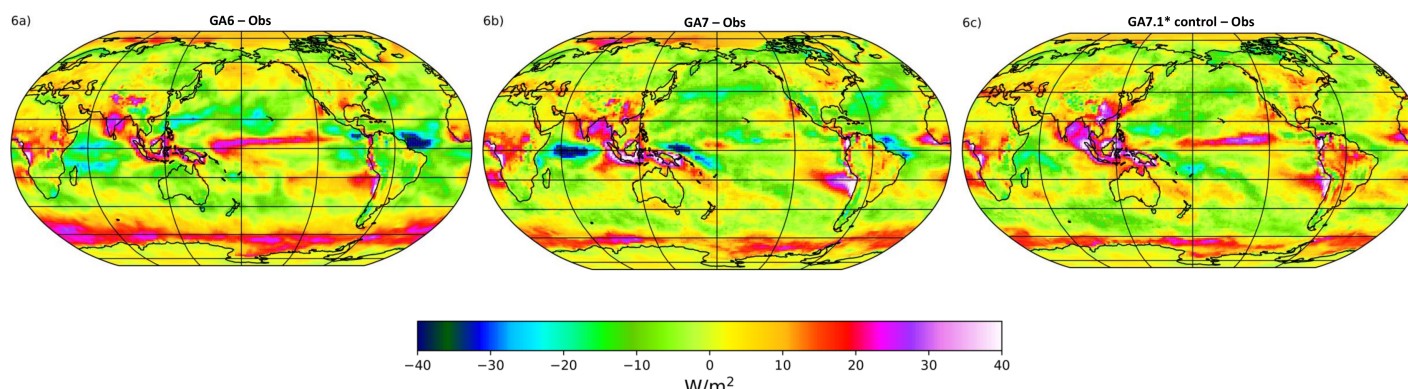

**Figure 6.** Spatial distribution of annual-mean SW CRE anomaly between model and observational data. (6a) SW CRE anomaly in an earlier model version, GA6 (Walters et al., 2017) (6b) Similar to (6a) but for the model version GA7 (Walters et al., 2019). (6c) Similar to (6a) but for the control model used in this study. Annual-mean for each model version is calculated from one year of daily mean data. Observational data is similar to the one used in Figure 3.



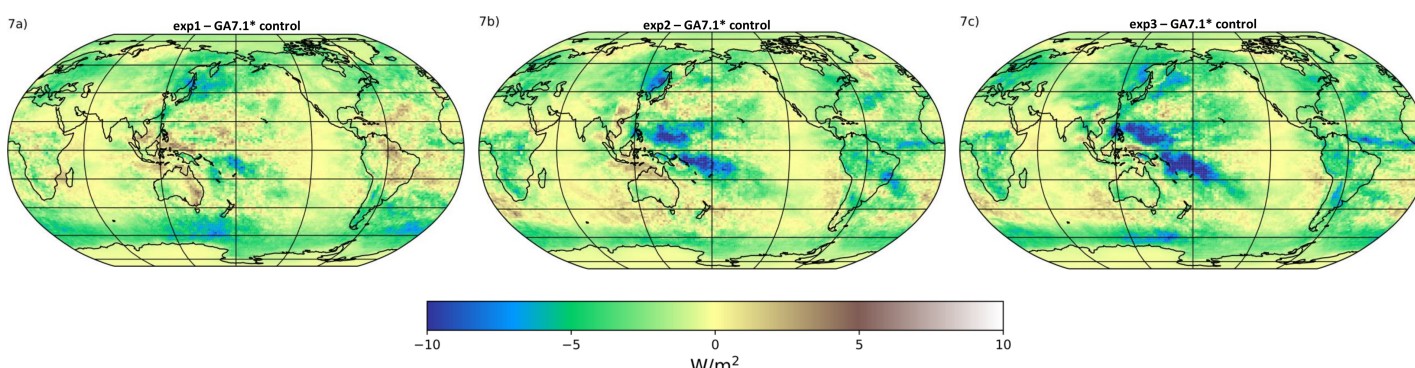

**Figure 7.** Spatial distribution of annual-mean SW CRE anomaly in different sensitivity experiments w.r.t the control run. (7a) exp1 - control, (7b) exp2 - control, (7c) exp3 - control. Annual-mean values are calculated from 12 hourly output over 20 years.