# Peer review of "Improving the Southern Ocean cloud albedo biases in a general circulation model"

_Atmospheric Chemistry and Physics, 2019_

## Referee Comment (RC1) · Anonymous Referee #1 · 30 Nov 2019

Summary:

Varma et al. describe the impact of changing the capacitance in the vapor deposition growth equation of ice crystals and changing the temperature that determines ice formation in the mixed-phase temperature regime for stratiform and convective clouds separately in a recent version of the Met Office's Unified Model. Their chosen capacitance is based on observations of a tropical anvil cloud and theoretical considerations. They analyze the impact of these sensitivity studies on radiative fluxes and ice as well as liquid water path. The focus of their analysis is on shortwave (SW) radiative fluxes in the Southern Hemisphere (SH) (50° S to 70° S).They find that the common bias of a too strong SW cloud radiative effect (CRE) in global climate models (GCMs) in this region is reduced in their sensitivity experiments.

[Figure]

General comment:

While the impact of changing the ice formation temperature on the Southern Ocean SW radiation bias has been shown in other studies, the idea to analyze the impact of changing the capacitance in the vapor deposition growth equation of ice crystals on the Southern Ocean SW radiation bias is novel and within the scope of Atmospheric Chemistry and Physics (ACP). But while the manuscript does show that changing the capacitance has an impact on the simulated SW radiation bias in the SH, the impact on longwave (LW) radiation and the Northern Hemisphere (NH) are not sufficiently considered (the latter is only shown in the supplementary material and not discussed), although the agreement with observations decreases. Furthermore, the properties of ice containing clouds will depend on further uncertain processes in a GCM like aggregation (efficiency). Even without cloud ice forming in mixed-phase temperature regime the SW radiation bias in the SH is not fully removed (exp2). It remains therefore unclear whether the claimed improved simulation of clouds over the Southern Ocean still holds when other aspects are considered. Therefore publication in ACP cannot be recommended unless these issues are addressed.

Specific comments:

P1L4: Is it more realistic to use the capacitance of Field et al. (2008) everywhere? In mixed-phase clouds (which occur frequently in the Southern ocean) riming can be important, hence more spherical ice particles can be present in these clouds.

P1L9: The reduction of the bias of $\sim$4 Wm-2 should be put into context of the strength of the bias in the model. Also this reduction in SW bias is accompanied by an increase in the LW bias.

P1L10-11: This is what Vergara-Temprado et al. (2018) have shown. What is your original contribution? In the conclusions it is written that INP's are not represented in the model, this can be mentioned in the abstract as well.

P1L19: You mean in the Southern Ocean. These studies include for example Williams et al. (2013) and Lohmann and Neubauer (2018).

P2L20: For which years is the sea-surface temperature climatology computed? Is Schuddeboom et al. (2019) the right citation for AMIP simulations?

P2L21: fig. 1 and further references to figures in the text: follow the manuscript preparation guidelines for authors of ACP.

P2L32: Why is the ventilation factor not considered in eq. (1)?

P3L7: For a sphere the capacitance is 0.5 x (maximum particle dimension).

P3L9: Morrison and Grabowski (2008) use the capacitance of a sphere for small spherical ice and 0.48 times the capacitance for a sphere for unrimed nonspherical crystals, and a linear interpolation in between for partially rimed crystals. Morrison and Milbrandt (2015) use the same in the predicted particle properties (P3) scheme. This is an even more realistic representation of ice crystal capacitance. Could this be implemented in the Unified Model?

P3L10-17: How is heterogeneous nucleation of ice represented in the model? Does it depend in ice nucleating particles (INP) concentrations or is it just a function of temperature (and if the latter, which function)? Not enough information is provided how heterogeneous and homogeneous freezing is implemented in the GA7.1*. What is the difference between the start-ice temperature and the all-ice temperature?

P3L20-21: Since this is not a model version that has been already described in another publication, it needs to be mentioned whether the experiment setup is similar to another study, otherwise details need to be given here. Why is 12 hourly output used? This means that the diurnal cycle is not well represented in the simulations. CERES-EBAF provides a diurnally complete representation of Earth's radiation budget (Loeb et al., 2018).

P3L22: ERA5 is a re-analysis dataset not an observational dataset.

P3L28-29: What was the reason to choose ERA5 as a reference for IWP given the large differences of IWP between different datasets (Duncan and Eriksson, 2018)? An uncertainty range for IWP should be added.

P3L31-32: Why are IWP/LWP shown only for these clouds? Shallow cumulus clouds may be interesting as well (Forbes et al., 2016).

P4L1-2: Why is the analysis split into this boundary layer types? Either provide a motivation and discussion for the different boundary layer types or remove this split.

P4L8 and all following occurrences: "w. r. t.": follow the manuscript preparation guidelines for authors of ACP.

P4L8-9: Why? Why does changing nucleation temperature not also impact LWP?

P5L6: What does "... at surface well" mean? Rephrase.

P5L13-14: That's an uncommon definition of SW CRE for model simulations. Typically two calls to the radiation routine are done, one with clouds and one without clouds. From these SW CRE is computed, taking into account cloud cover. How is SW CRE computed in partly cloudy gridboxes?

P5L16-18: Why do exp1 and exp3 show a stronger reduction in SW CRE than exp2. In exp2 the least cloud ice should be present in the mixed-phase clouds so why is the SW CRE larger in exp2?

P5L31-32: Why (see previous comment)?

P5L33: What are "eastern sects"?

P6L1: Provide references for this statement.

P6L10: Why are low INP concentrations relevant? Are INP's used in any of the experiments?

P6L11: "temperatures between the homogeneous and the heterogeneous freezing

points"; rewrite, it's unclear what is meant

P6L17: 0.5 x d is used above

P6L19-20: Why are these then shown?

P6L22: This is not discussed anywhere. Either a discussion is added or the respective experiment and its results should be removed.

P6L29-30: Is there an explanation why the capacitance change has no significant impact in the tropics?

P6L29-30: Why does the capacitance change not significantly change or even decrease SW CRE in the tropics? Is this model dependent?

P7L9: There's no discussion why these temperature thresholds have been chosen for the sensitivity experiments. Are this thresholds considered to be realistic?

P7L20-21: As these changes are not described in the literature or publicly accessible, they need to be described here.

P7L25: The link is not publicly accessible.

Table 1: The experiments could have more meaningful names which indicate what has been changed. Why is the all-ice temperature 1° C larger than the start-ice temperature?

Fig. 1: the sensitivity experiments should be added to this figure. ERA5 is a re-analysis dataset not an observational dataset.

Fig. 3 and all similar figures: a vertical line at 0 Wm-2 is missing. Also these figures make it hard to compare different experiments. One panel should rather show anomalies for one variable but for all experiments and observations. Where do the sensible and latent heat observations come from?

Fig. S8b shows that exp2 still has a SW top-of-the-atmosphere (TOA) bias although

no more ice is present in the mixed-phase temperature range. This indicates that the SW TOA bias is not only due to the wrong phase of mixed-phase clouds in GA7.1* but that there are biases also in other clouds.

Fig. S8 shows that the SW TOA bias in the NH increases in exp1 compared to ctrl. Also from 50° S to 60° S the SW TOA bias increases in exp1 compared to ctrl. Is changing the capacitance really improving the agreement with observations? The root-mean-square error and correlation coefficient with respect to CERES would show if the experiments are an improvement globally.

References:

Duncan, D. I. and Eriksson, P.: An update on global atmospheric ice estimates from satellite observations and reanalyses, Atmos. Chem. Phys., 18, 11205–11219, https://doi.org/10.5194/acp-18-11205-2018, 2018.

Forbes, R, Geer, AJ, Lonitz, K, Ahlgrimm, M.: Reducing systematic errors in cold-air outbreaks, ECMWF Newsletter 146, https://doi.org/ 10.21957/s41h7q7l, 2016.

Lohmann, U. and Neubauer, D.: The importance of mixed-phase and ice clouds for climate sensitivity in the global aerosol–climate model ECHAM6-HAM2, Atmos. Chem. Phys., 18, 8807–8828, https://doi.org/10.5194/acp-18-8807-2018, 2018.

Morrison, H. and Grabowski, W. W.: A novel approach for representing ice microphysics in models: Description and tests using a kinematic framework. J. Atmos. Sci., 65, 1528–1548, doi:10.1175/ 2007JAS2491.1, 2008.

Morrison, H. and Milbrandt, J. A.: Parameterization of Cloud Microphysics Based on the Prediction of Bulk Ice Particle Properties, Part I: Scheme Description and Idealized Tests, J. Atmos. Sci., 72, 287–311, https://doi.org/10.1175/JAS-D-14-0065.1, 2015.

Williams, K. D., Bodas-Salcedo, A., Déqué, M., Fermepin, S., Medeiros, B., Watanabe, M., Jakob, C., Klein, S. A., Senior, C. A., and Williamson, D. L.: The Transpose-AMIP II Experiment and Its Application to the Understanding of Southern Ocean Cloud Biases

in Climate Models, J. Climate, 26, 3258–3274, 2013.

---

## Referee Comment (RC2) · Anonymous Referee #2 · 3 Dec 2019

General Comments

Varma et al. perform global climate model (GCM) simulations with modified cloud parameterizations to investigate why GCMs underestimate cloud albedo over the Southern Ocean. This is a major and long-standing bias in GCMs. The authors investigate the hypothesis that albedo bias over the Southern Ocean is caused by an overly simplified treatment of ice-crystal shape that is used by current cloud parameterizations. The question that the authors investigate is important and fits within the scope of Atmospheric Chemistry and Physics. However, the study has some issues involving justification of the experimental design, discussion of the simulations, and clarity of the figures and writing. If these issues are addressed, then the manuscript might be acceptable for publication. I therefore recommend major revision.

[Figure]

**Specific Comments**

**Title**

I think "improved" is not appropriate to use in the title since the authors did not improve the theory on which the cloud parameterizations are based. Changing the tuning parameters in a model, as the authors have done in this study, is not the same thing as improving the model. I suggest that the title be changed to something like "Bias of Southern Ocean cloud albedo in a general circulation model linked to ice-crystal shape."

**Abstract**

All of the abstract is fine except for the last sentence. The last sentence should be removed because the authors did not do any new work to justify this statement ("We hypothesize that such abundant supercooled liquid cloud is the result of a paucity of ice nucleating particles in this part of the atmosphere."). It is unethical to make this statement in the abstract because the statement is based entirely on the work of others. It would be fine to include this statement in the discussion section with proper references, of course.

**Data and Experimental Set-up**

The experimental design needs to be explained and justified in more detail. For instance, the authors perform a sensitivity study in which the ice-crystal shape is modified. This is done by multiplying the "capacitance" (C) value by a factor of 0.5, which effectively changes the ice-crystal shape from spheres to ellipsoids. However, the authors do not cite any theoretical or observational work to justify their choice of 0.5 until the Discussion section (pg. 6 line 8), and even there it is simply stated that the choice of C is reasonable without any explanation. More background information justifying the choice of C=0.5 is needed in Section 2.1. It would also be nice if the authors provided some justification for their choice that is based on in situ observations over

the Southern Ocean, perhaps from the recent SOCRATES field campaign. A second issue is that, as far as I can tell, some of the simulations and discussion are unrelated to the study goals. Simulations exp2 and exp3 use modified temperatures for ice nucleation in the convection and microphysics parameterizations. How do these experiments contribute to the goal of understanding how ice-crystal shape affects Southern Ocean cloud albedo? Also, the control simulation is compared to older versions of the model with no explanation of how this comparison helps to understand the cause of the cloud albedo bias in the current model (pg. 5 line 20, Figure 6). I do not understand the value of exp2, exp3, or the older versions of the model presented in Figure 6. Please discuss this or remove the content.

Results and Discussion

The Results section is hard to follow. It would help to organize the figures and text in a consistent way. The text discusses model bias in the TOA and surface energy budget terms one at a time, so it would be helpful if the data presented in Figure 3-5 were also organized based on different energy budget terms. For instance, Figure 3 could have one panel that shows LW TOA in ctrl, exp1, exp2, and exp3; another panel that shows SW TOA in ctrl, exp1, exp2, and exp3; and so on. Since model bias is the quantity of interest, it would also help to show all of the anomalies relative to observed values (e.g. ctrl – obs, exp1 – obs, exp2 – obs, exp3 – obs) rather than anomalies relative to the ctrl experiment in some of the panels and anomalies relative to observations in other panels. Another issue is that the content of the Discussion section doesn't seem to logically follow from the content of the Results section. The Results section describes how the model biases change as a result of the modifications to the cloud parameterizations, which is fine. But no clear conclusion about what was learned from these simulations is reached in the Discussion section. Should other modeling groups change the ice crystal shape in their models? If so, what range of capacitance values is suggested by observations and theory, and what values do the authors recommend using? How much of the Southern Ocean cloud albedo bias will be fixed by changing

the ice-crystal shape? Please make a clear statement about what was learned from your work before starting the discussion about how other studies say that ice-nucleating particles are the critical thing to study (pg. 6 line 31).

Technical Corrections

Figure 1 – Change axis label to "IWP (km/m2)" to match the rest of the text.

Figure 2 – Why is the range of the x-axis so much larger in 2a-b than in 2c-d? Make the axis range consistent across all panels.

Figure 2 – I suggest moving all of the information about cloud types from the figure caption to the main text.

Figure 3,4 – Please organize the data so that one panel shows one energy budget term only, and that all anomalies are shown relative to observations, as mentioned in my comments on "Results and Discussion" above.

Figure 6 – What value does this figure add to the study? I think this figure should be removed

Figure 7 – What does this figure show that isn't already shown in Figure 5? It shows a big response in the tropical western Pacific to changing the nucleation temperature, but this isn't relevant to understanding Southern Ocean cloud albedo biases.

Figure 6,7 – The colorbar makes these figures very difficult to read. Please change the colorbar to a two-color scale with white at zero. For example, the colorbar could have red for positive values, white for near-zero values, and blue for negative values.

Pg. 1 line 19 "observed radiation biases" – delete "observed"

Pg. 2 line 6 – specify that "this model problem" means cloud albedo bias over the Southern Ocean

Pg. 2 line 8 – I recommend moving the sentence "In the present study, we

investigate. . ." to the end of the preceding paragraph and moving the sentence "Here, we define a SO. . ." to Section 2 Data and experimental set-up. I think it helps to finish the Introduction with a concise statement of the study goals, which is what the first sentence does.

Pg. 2 line 12 – Why isn't this paragraph in section 2.1 Model set-up?

Pg. 2 line 13 – Is it necessary to put the model description in an appendix? Appendix A is only one paragraph long, after all. It improves the clarity of the paper if the reader doesn't have to jump around between different sections.

Pg. 3 line 14 "parametrised convection scheme" – "parametrised" is redundant and can be deleted.

Pg. 4 line 8 – Why does modifying the capacitance value affect liquid and ice? Does the capacitance value control the diffusional growth of liquid droplets as well? If so, then I don't think that C=0.5 is realistic for liquid droplets. Also, why does changing the ice nucleation temperature predominantly affect IWP? I think other studies suggest that it should affect both LWP and IWP [e.g. Kay et al., 2016].

Pg. 4 line 11, Pg. 5 line 9, Pg. 5 line 19 – Please don't just state that these figures are included in the supporting information. You need to say what the figures show and how they contribute to the findings of the study.

Pg. 4 line 17 – Why is the change in TOA LW flux so large in your simulations? LW radiation was not part of the motivation, yet TOA LW flux is more sensitive than TOA SW flux to the model modifications made in this study. Please explain this.

Pg. 4 line 22 – By "show an increase" do you mean an increase relative to the control experiment? Please clarify.

Pg. 5 line 1 – It would help to discuss the difference between the control simulation and observations first to establish the baseline model bias, then discuss how the bias changes in exp1-3. Please rearrange content accordingly.

[Figure]

Pg. 6 line 18 "The atmosphere-only model studied here does perform better..." – Please use more specific language. For example, "model bias in SW CRE is reduced over the Southern Ocean."

---

## Author Comment (AC1) · 29 Apr 2020

**Reviewer 1 comments**

We would like to thank the anonymous reviewer for his/her comments. Below is the detailed point-by-point reply to the comments.

**1   Summary**

"But while the manuscript does show that changing the capacitance has an impact on the simulated SW radiation bias in the SH, the impact on longwave (LW) radiation and the Northern Hemisphere (NH) are not sufficiently considered (the latter is only shown in the supplementary material and not discussed), although the agreement with observations decreases. Furthermore, the properties of ice containing clouds will depend on further uncertain processes in a GCM like aggregation (efficiency). Even without cloud ice forming in mixed-phase temperature regime the SW radiation bias in the SH is not fully removed (exp2). It remains therefore unclear whether the claimed improved simulation of clouds over the Southern Ocean still holds when other aspects are considered. Therefore publication in ACP cannot be recommended unless these issues are addressed."

We have modified the manuscript as per the points below.

**2  Specific Points**

1. P1L4: Is it more realistic to use the capacitance of Field et al. (2008) everywhere? In mixed-phase clouds (which occur frequently in the Southern ocean) riming can be important, hence more spherical ice particles can be present in these clouds.

Riming would eventually produce quasi-spherical particles, but in highly supercooled environments at water saturation, the growth due to deposition is likely to be faster. This is not only due to the high ice supersaturation but also because riming with cloud droplets is diminished. Observation by Harimaya (1975) shows that droplets with diameters less than 10 $\mu$m are too small to be collected onto ice crystals. Westbrook and Illingworth (2013) show some examples of pristine particles in supercooled layer clouds. Particles like these stellars have a capacitance close to the one we are using. Also the description of their fig5 points out the strong $Z_{DR}$ (differential reflectivity) signal indicating non-spherical oblate particles.

References:

Westbrook, C. and Illingworth, A.: The formation of ice in a longlived supercooled layer cloud, Q. J. Roy. Meteor. Soc, 139, 2209–2221, doi:10.1002/qj.2096, 2013.

Harimaya, T., 1975: The riming properties of snow crystals. J. Meteor. Soc. Japan, 53, 384–392.

[Figure]

2. P1L9: The reduction of the bias of $\sim$4 Wm$^{-2}$ should be put into context of the strength of the bias in the model. Also this reduction in SW bias is accompanied by an

increase in the LW bias.

The reduction of SW cloud radiative effect of 4 W/m2 that we have shown is for the TOA since changes are more certain compared to the surface flux. As mentioned in the revised 'Observational data' section, the surface changes are prone to more uncertainties. CERES surface data itself depends on the uncertainties in the radiative transfer model.
Page 4; Section 2.2

3. P1L10-11: This is what Vergara-Temprado et al. (2018) have shown. What is your original contribution? In the conclusions it is written that INP's are not represented in the model, this can be mentioned in the abstract as well.

Removed the last sentence of abstract in the new version that mentions INP

4. P1L19: You mean in the Southern Ocean. These studies include for example Williams et al. (2013) and Lohmann and Neubauer (2018).

References added.
Page 1; Section 1

5. P2L20: For which years is the sea-surface temperature climatology computed? Is Schuddeboom et al. (2019) the right citation for AMIP simulations?

Reynolds SST for the years 1981 – 2012 has been used. Gates et al 1999 is the reference for AMIP (reference already included). Schuddeboom et al., 2019 is another study where a version of the control run used in of our study was used, hence cited it. Removed it to avoid any confusion.
Page 2; Section 2.1

6. P2L21: fig. 1 and further references to figures in the text: follow the manuscript preparation guidelines for authors of ACP

Corrected in the new version

7. P2L32: Why is the ventilation factor not considered in eq. (1)?

The capacitance value of 0.5 that we have used in our model does take into account ventilation factor as well (as per Field et al 2008) although it is challenging to quantify the effects of ventilation factor and deposition rates separately.

8. P3L7: For a sphere the capacitance is 0.5 x (maximum particle dimension). This is mostly difference in the naming convention or defining the 'capacitance'. Wesbrook et al., 2008 or Field et al., 2008 defines C = 0.5 for spheres and the product C*D is the capacitance. It is however conveying the same message. Essentially the default value of 'capacitance' is reduced from 1xd to 0.5xd in our study.

9. P3L9: Morrison and Grabowski (2008) use the capacitance of a sphere for small spherical ice and 0.48 times the capacitance for a sphere for unrimed nonspherical crystals, and a linear interpolation in between for partially rimed crystals. Morrison and Milbrandt (2015) use the same in the predicted particle properties (P3) scheme. This is an even more realistic representation of ice crystal capacitance. Could this be implemented in the Unified Model?

No, this approach cannot be applied in the Unified Model as we don't have a prognostic for riming fraction on ice like in their study.

10. P3L10-17: How is heterogeneous nucleation of ice represented in the model?

Does it depend in ice nucleating particles (INP) concentrations or is it just a function of temperature (and if the latter, which function)? Not enough information is provided how heterogeneous and homogeneous freezing is implemented in the GA7.1*. What is the difference between the start-ice temperature and the all-ice temperature?

The control model does not have any ice-nuclei dependency for the heterogeneous nucleation. The heterogeneous nucleation temperature is simply following the temperature dependent function suggested by Fletcher [1962] (N. H. Fletcher. The physics of rainclouds. Cambridge University Press, London, UK, 1962). This then gets multiplied by a small 'seed' ice content for ice free clouds in order that the other micro physical terms can grow it. As far as homogeneous nucleation of liquid water is concerned, all liquid water at temperatures less than $-40°$C is instantaneously frozen to form ice particles according to Rogers and Yau [1989]. (R. R. Rogers and M. K. Yau. A short course in cloud physics. Pergamon Press, Oxford, 3rd edition, 1989). At 'start-ice temperature', the detraining of liquid condensate as ice begins in the model and by 'all-ice temperature', all condensate is detrained as ice. These details have been added in the revised version under Model set-up.
Page 3; line 12 - 29

11. P3L20-21: Since this is not a model version that has been already described in another publication, it needs to be mentioned whether the experiment setup is similar to another study, otherwise details need to be given here. Why is 12 hourly output used? This means that the diurnal cycle is not well represented in the simulations. CERES-EBAF provides a diurnally complete representation of Earth's radiation budget (Loeb et al., 2018).

Appendix modified (Page 8). In the newer version of the manuscript, we now use the daily-mean values for radiative fluxes from model

12. P3L22: ERA5 is a re-analysis dataset not an observational dataset.

We have removed ERA5 data in the new version. a modified Observational data section is included.
Page 4; Section 2.2 lines 7 - 12

13. P3L28-29: What was the reason to choose ERA5 as a reference for IWP given the large differences of IWP between different datasets (Duncan and Eriksson, 2018)? An uncertainty range for IWP should be added.

In the newer version, we are not using this comparison for IWP.

14. P3L31-32: Why are IWP/LWP shown only for these clouds? Shallow cumulus clouds may be interesting as well (Forbes et al., 2016).

The focus of this study is mostly on the stratocumulus boundary layer type clouds. Hence, we chose the corresponding types as mentioned in the main material of the manuscript. We have added analyses for other boundary layer types in the Supplementary material. A brief description of other types has now been included in the main material as well.
Page 4 ; lines 17-23

15. P4L1-2: Why is the analysis split into this boundary layer types? Either provide a motivation and discussion for the different boundary layer types or remove this split.

Similar to previous comment. More details included now.

16. P4L8 and all following occurrences: "w. r. t.": follow the manuscript preparation

guidelines for authors of ACP.

Corrected

17. P4L8-9: Why? Why does changing nucleation temperature not also impact LWP?

The zonal mean liquid water paths that we have shown here are dominated by the fronts mostly. So, even if ice nucleation temperature shows some sensitivity, they are mostly away from the frontal systems and mostly restricted to the shallow boundary layer types.
Page 4 ; line 30

18. P5L6: What does ". . . at surface well" mean? Rephrase.

Corrected

19. P5L13-14: That's an uncommon definition of SW CRE for model simulations. Typically two calls to the radiation routine are done, one with clouds and one without clouds. From these SW CRE is computed, taking into account cloud cover. How is SW CRE computed in partly cloudy gridboxes?

We have rephrased the sentence. In the model, for each grid box there is a cloudy and non-cloudy flux. From these fluxes, the CRE can be calculated using the amount of cloud fraction.
Page 5 line 32

20. P5L16-18: Why do exp1 and exp3 show a stronger reduction in SW CRE than exp2. In exp2 the least cloud ice should be present in the mixed-phase clouds so why

is the SW CRE larger in exp2?

We do acknowledge some uncertainties in the effect of nucleation temperature on fluxes. There are some detrimental effects due to changes in the nucleation temperature that could be mostly due to the changes in the vertical distribution of the clouds affecting not just the low clouds but also the high clouds. By changing the nucleation temperature, we are essentially modifying the level at which freezing occurs. So, when we don't freeze the water lower down then it can go higher up in the atmosphere probably creating cirrus clouds and thus change the high cloud characteristics (thus affecting both short/long waves). A detailed examination of the effects on fluxes is not intended to be within the scope of this study. However, we do acknowledge the importance of this aspect and have incorporated that to be continued in future work. We have stressed this point and made more clarity in the discussion and conclusion sections. Page 6 ; lines 20-25 and Page 7 ; lines 23-25

21. P5L31-32: Why (see previous comment)?

Reply similar to the previous one

22. P5L33: What are "eastern sects"?

Removed

23. P6L1: Provide references for this statement.

Added ; Page 7 lines 1-4

24. Why are low INP concentrations relevant? Are INP's used in any of the experiments?

Removed the sentence and modified Discussion section; Page 7

25. "temperatures between the homogeneous and the heterogeneous freezing points"; rewrite, it's unclear what is meant

Removed the sentence

26. P6L17: 0.5 x d is used above

Similar to comment 8

27. P6L19-20: Why are these then shown?

We have modified the Results and Discussion sections

28. P6L22: This is not discussed anywhere. Either a discussion is added or the respective experiment and its results should be removed.

Added details in Page 3 lines 12-33

29. P6L29-30: Is there an explanation why the capacitance change has no significant impact in the tropics?

In an earlier study by Furtado et al., 2016 (using the NWP model), it has been shown that for tropics and subtropics there is a general tendency by the model to overpredict the LWP in response to microphysics modifications. Increasing the stratiform cloud LWP will cause more SW radiation to be reflected back to space. But over the Southern Ocean, this effect is beneficial because the Unified Model has a large negative bias in outgoing SW radiation in that region. Some possible reasons mentioned are that of flaws in parametrizations, uncertainities in the estimation of LWP in the convection scheme etc. Basically, in a frontal steady state, the capacitance doesn't have much of an impact compared to more dynamic sites like that of super cooled liquid water clouds. Further details can be found in Furtado et al., 2016.
Page 6 lines 11-19

30. P6L29-30: Why does the capacitance change not significantly change or even decrease SW CRE in the tropics? Is this model dependent?

For the first part of the comment, response similar to the previous point. For the second part, mostly it is not model dependent. Changes in capacitance can be translated to any model that uses that factor. Basically any model that is using capacitance change is a sink of water vapour. The impact of capacitance predominantly depends on the amount liquid water already available in the model. So, if some models have very less liquid water, then the impact of capacitance might not be much.

31. P7L9: There's no discussion why these temperature thresholds have been chosen for the sensitivity experiments. Are this thresholds considered to be realistic?

We have added more details in the text; Page 3 lines 20-22

32. P7L20-21: As these changes are not described in the literature or publicly accessible, they need to be described here.

Appendix modified

33. P7L25: The link is not publicly accessible.

As it is not a published version for the control, we have added some additions to its predecessor that are relevant to this study in the Appendix.

34. Table 1: The experiments could have more meaningful names which indicate what has been changed. Why is the all-ice temperature 1 deg C larger than the start-ice temperature?

We have renamed the experiment names for clarity. The 1 deg C change is merely technical to avoid division by zero in the code.

35. Fig. 1: the sensitivity experiments should be added to this figure. ERA5 is a re-analysis dataset not an observational dataset.

We have removed this figure in the modified version

36. Fig. 3 and all similar figures: a vertical line at 0 Wm-2 is missing. Also these figures make it hard to compare different experiments. One panel should rather show anomalies for one variable but for all experiments and observations. Where do the sensible and latent heat observations come from?

We have included modified figures. Observational Data section 2.2 has also been modified to accommodate the changes.

37. Fig. S8b shows that exp2 still has a SW top-of-the-atmosphere (TOA) bias although no more ice is present in the mixed-phase temperature range. This indicates that the SW TOA bias is not only due to the wrong phase of mixed-phase clouds in

GA7.1* but that there are biases also in other clouds.

We have removed this figure. New figures using observational data is used in the main text.

38. Fig. S8 shows that the SW TOA bias in the NH increases in exp1 compared to ctrl. Also from 50S to 60S the SW TOA bias increases in exp1 compared to ctrl. Is changing the capacitance really improving the agreement with observations? The rootmean-square error and correlation coefficient with respect to CERES would show if the experiments are an improvement globally.

We have now removed this figure. As we have now noted in the revised main text, change in capacitance is mostly favorable for the dynamic regions like super cooled liquid clouds (SO for instance). Since, the focus of our study is mainly SO, we have included fluxes and SW CRE plots (zonal) only for the SH. The global spatial plot is shown mostly for completeness and also as a motivation for the importance of INP and we have emphasized this is in the newer version.

---

## Author Comment (AC2) · 29 Apr 2020

**Reviewer 2 comments**

We would like to thank the anonymous reviewer for his/her comments. Below is the detailed point-by-point reply to the comments.

**1  Summary**

"However, the study has some issues involving justification of the experimental design, discussion of the simulations, and clarity of the figures and writing. If these issues are addressed, then the manuscript might be acceptable for publication. I therefore recommend major revision."

Manuscript modified as per comments below

**2  Specific Points**

1.Title : I think "improved" is not appropriate to use in the title since the authors did not improve the theory on which the cloud parameterizations are based. Changing

the tuning parameters in a model, as the authors have done in this study, is not the same thing as improving the model. I suggest that the title be changed to something like "Bias of Southern Ocean cloud albedo in a general circulation model linked to ice-crystal shape."

Title modified

2.Abstract : All of the abstract is fine except for the last sentence. The last sentence should be removed because the authors did not do any new work to justify this statement ("We hypothesize that such abundant supercooled liquid cloud is the result of a paucity of ice nucleating particles in this part of the atmosphere."). It is unethical to make this statement in the abstract because the statement is based entirely on the work of others. It would be fine to include this statement in the discussion section with proper references, of course.

Modified

3. Data and Experimental Set-up: The experimental design needs to be explained and justified in more detail. For instance, the authors perform a sensitivity study in which the ice-crystal shape is modified. This is done by multiplying the "capacitance" (C) value by a factor of 0.5, which effectively changes the ice-crystal shape from spheres to ellipsoids. However, the authors do not cite any theoretical or observational work to justify their choice of 0.5 until the Discussion section (pg. 6 line 8), and even there it is simply stated that the choice of C is reasonable without any explanation. More background information justifying the choice of C=0.5 is needed in Section 2.1. It would also be nice if the authors provided some justification for their choice that is based on in situ observations over the Southern Ocean, perhaps from the recent SOCRATES field campaign.

We have added some more information in Section 2.1 (Page 3 lines 5-8). Regarding in situ observations, we are not aware of any capacitance or aspect ratio of ice crystals related data from SOCRATES.

A second issue is that, as far as I can tell, some of the simulations and discussion are unrelated to the study goals. Simulations exp2 and exp3 use modified temperatures for ice nucleation in the convection and microphysics parameterizations. How do these experiments contribute to the goal of understanding how ice-crystal shape affects Southern Ocean cloud albedo?

We have added more details regarding this in Section 2.1 lines 5-29

Also, the control simulation is compared to older versions of the model with no explanation of how this comparison helps to understand the cause of the cloud albedo bias in the current model (pg. 5 line 20, Figure 6). I do not understand the value of exp2, exp3, or the older versions of the model presented in Figure 6. Please discuss this or remove the content.

We have now removed the comparison with earlier model versions

4. Results and Discussion
The Results section is hard to follow. It would help to organize the figures and text in a consistent way. The text discusses model bias in the TOA and surface energy budget terms one at a time, so it would be helpful if the data presented in Figure 3-5 were also organized based on different energy budget terms. For instance, Figure 3 could have one panel that shows LW TOA in ctrl, exp1, exp2, and exp3; another panel that shows SW TOA in ctrl, exp1, exp2, and exp3; and so on. Since model bias is the quantity of interest, it would also help to show all of the anomalies relative to observed values (e.g. ctrl − obs, exp1 − obs, exp2 − obs, exp3 − obs) rather than anomalies relative

to the ctrl experiment in some of the panels and anomalies relative to observations in other panels.

We have added modified figures

Another issue is that the content of the Discussion section doesn't seem to logically follow from the content of the Results section. The Results section describes how the model biases change as a result of the modifications to the cloud parameterizations, which is fine. But no clear conclusion about what was learned from these simulations is reached in the Discussion section. Should other modeling groups change the ice crystal shape in their models? If so, what range of capacitance values is suggested by observations and theory, and what values do the authors recommend using? How much of the Southern Ocean cloud albedo bias will be fixed by changing the ice-crystal shape? Please make a clear statement about what was learned from your work before starting the discussion about how other studies say that ice-nucleating particles are the critical thing to study (pg. 6 line 31).

Modified the Results, Discussion and Conclusion sections to make our findings more clear

5. Technical Corrections
Figure 1 – Change axis label to "IWP (km/m2)" to match the rest of the text.

Removed the figure

Figure 2 – Why is the range of the x-axis so much larger in 2a-b than in 2c-d? Make the axis range consistent across all panels.

Modified

Figure 2 – I suggest moving all of the information about cloud types from the figure caption to the main text.

Added cloud type details in the main text; Page 4 lines 20-23

Figure 3,4 – Please organize the data so that one panel shows one energy budget term only, and that all anomalies are shown relative to observations, as mentioned in my comments on "Results and Discussion" above.

Modified figures added

Figure 6 – What value does this figure add to the study? I think this figure should be removed

This figure has been removed.

Figure 7 – What does this figure show that isn't already shown in Figure 5? It shows a big response in the tropical western Pacific to changing the nucleation temperature, but this isn't relevant to understanding Southern Ocean cloud albedo biases.

We have added more details in the results section explaining this.

Figure 6,7 – The colorbar makes these figures very difficult to read. Please change the colorbar to a two-color scale with white at zero. For example, the colorbar could have red for positive values, white for near-zero values, and blue for negative values.

Modified figure added

Pg. 1 line 19 "observed radiation biases" – delete "observed"

Deleted

Pg. 2 line 6 – specify that "this model problem" means cloud albedo bias over the Southern Ocean

Modified

Pg. 2 line 8 – I recommend moving the sentence "In the present study, we investigate. . ." to the end of the preceding paragraph and moving the sentence "Here, we define a SO. . ." to Section 2 Data and experimental set-up. I think it helps to finish the Introduction with a concise statement of the study goals, which is what the first sentence does.

Modified; Page 2 lines 7-10

Pg. 2 line 12 – Why isn't this paragraph in section 2.1 Model set-up?

We have added few more background details in the UM model version in the Appendix.

Pg. 2 line 13 – Is it necessary to put the model description in an appendix? Appendix A is only one paragraph long, after all. It improves the clarity of the paper if the reader doesn't have to jump around between different sections.

We have modified Section 2. Appendix is included with more details now as these are not publicly accessible yet due to licensing issue.

Pg. 3 line 14 "parametrised convection scheme" – "parametrised" is redundant and can be deleted.

Modified

Pg. 4 line 8 – Why does modifying the capacitance value affect liquid and ice? Does the capacitance value control the diffusional growth of liquid droplets as well? If so, then I don't think that C=0.5 is realistic for liquid droplets. Also, why does changing the ice nucleation temperature predominantly affect IWP? I think other studies suggest that it should affect both LWP and IWP [e.g. Kay et al., 2016].

The effect of capacitance on liquid is mostly indirect. When ice grows slower, it leaves more water vapor around to condense to liquid drops. And if the capacitance is high, then ice crystals grow faster and there is less liquid. So, by making the capacitance value to 0.5 from the default value of 1.0, we are in a way reducing the depositional

growth of ice crystals, leaving more room for water vapor to condense (e.g.Wegener, 1911; Bergeron, 1935; Findeisen : same also provided in the main text reference). Kay et al., 2016 shows the improvement in radiation biases over SO by modifying the shallow convection temperature rather than tuning the cloud microphysics.

Pg. 4 line 11, Pg. 5 line 9, Pg. 5 line 19 – Please don't just state that these figures are included in the supporting information. You need to say what the figures show and how they contribute to the findings of the study.

We have modified the Supplementary section and its reference in the main text. Page 4 lines 22-24

Pg. 4 line 17 – Why is the change in TOA LW flux so large in your simulations? LW radiation was not part of the motivation, yet TOA LW flux is more sensitive than TOA SW flux to the model modifications made in this study. Please explain this.

For the atmosphere only version of the model (i.e. without any interactive sea surface temperature), the LW changes are slightly complicated because any changes in the radiation budget of the SW does not have any impact on the outgoing radiation from the sea-surface. But the changes that we see here in the LW could be mostly due to the changes associated with the amount of cloud cover and cloud height that we observe in the experiments. When there is more horizontal cloud cover then more of the surface is covered and that will have an impact on the LW distribution. Also, when the cloud height changes i.e when it becomes thicker that could also impact the LW. We have now emphasized in the discussion/conclusions sections that the capacitance changes are aimed mostly at the boundary layer clouds and nucleation temperature changes could also influence the high clouds. We have also now made it more clear

that it is the SW flux that is mostly benefiting and also mention about the detrimental effects on other fluxes.
Page 6 lines 20-25

Pg. 4 line 22 – By "show an increase" do you mean an increase relative to the control experiment? Please clarify.

We have modified the Results section.

Pg. 5 line 1 – It would help to discuss the difference between the control simulation and observations first to establish the baseline model bias, then discuss how the bias changes in exp1-3. Please rearrange content accordingly.

Modified

Pg. 6 line 18 "The atmosphere-only model studied here does perform better. . ." – Please use more specific language. For example, "model bias in SW CRE is reduced over the Southern Ocean."

Modified

---

## Referee Report (RR1)

**Review of "Improved simulation of clouds over the Southern Ocean in a General Circulation Model" by Vidya Varma et al.**
*Atmospheric Chemistry and Physics* **Manuscript # acp-2019-884**

**General Comments**

Varma et al. perform global climate model (GCM) simulations with modified cloud parameterizations to investigate why GCMs underestimate cloud albedo over the Southern Ocean. This is a long-standing bias in GCMs, so it is an important scientific problem. The results are important, clearly presented, and fit within the scope of *Atmospheric Chemistry and Physics*. I have several suggestions for minor technical corrections that are listed below. I recommend that the paper be accepted for publication after these issues are addressed.

**Technical Corrections**

- Pg. 3 line 24 – Change "changing the detrainment temperatures to be very cold from the default values" to "changing the detrainment temperatures to be colder than the default values."
- Pg. 4 line 32 – When you say "changes to nucleation temperature shows some sensitivity" it is unclear what you are talking about. Do you mean LWP shows some sensitivity to changes in nucleation temperature? Please be more specific.
- Pg. 5 line 22 & 25 – change "is showing" and "showed" to "show"
- Pg. 5 line 33 – Change "taking the anomaly of TOA SW flux between the cloudy and non-cloudy grid boxes" to "taking the difference between all-sky and clear-sky conditions."
- Pg. 6 line 11 – Change "don't" to "do not"
- Pg. 7 line 3 – Do you really need to use the abbreviation "SLC"? You only use this word three times in the paper. The abbreviation SLC is defined in the introduction and then used near the end of the paper, so some readers may forget what it means by the time they see it in the discussion section. I suggest that you spell out the word and drop the abbreviation.
- Pg. 7 line 18 – What is a "frontal steady state"? Can you define this more clearly?
- Conclusion – It would help to state how much of a reduction you see in the Southern Ocean shortwave radiation bias in your simulations, rather than just saying that the bias is reduced.
- Figure 5 – Make the scale of the x-axis the same for 5a and 5b.

---

## Referee Report (RR2)

**Review of „Reducing the Southern Ocean cloud albedo biases in a general circulation model" by Varma, V., Morgenstern, O., Field, P., Furtado, K., Williams, J., and Hyder, P.**

General comment:
The authors improved the representation of their results and clarified many concerns. However, several issues remain. In particular it is unclear if the new shape parameter is indeed an improvement as the changes in SW fluxes in the SO region in the experiments are accompanied by compensating changes in LW fluxes and undesirable changes in NH SW fluxes. Also the biases need to be quantified globally and regionally. From the figures it is unclear if there is a reduction in the Southern Ocean cloud albedo bias. Since the new shape parameter is applied globally also the global impact needs to be shown. A table with global and regional (50° S to 70° S) mean values should be added to quantify the changes by the experiments. This should at least contain the bias to observations but also the correlation coefficient and root mean square error for SW and LW TOA fluxes (at least). For example abstract and main text suggest that not enough SW radiation is reflected in the SO region (50° S to 70° S) in the ctrl experiment compared to observations, but is this actually the case? This bias needs to be quantified. Furthermore, no clear recommendation is given where the new shape parameter should be applied (globally or only in certain regions). Publication in ACP can only be recommended after these issues are addressed.

Specific comments:
Title: Is the Southern Ocean cloud albedo bias reduced compared to observations? This needs to be quantified, see general comment.

P1L1-10: The SW changes are accompanied by LW changes; this needs to be mentioned in the abstract and main text and also be discussed in the main text.

P2L18: Unless this a named sea surface temperature climatology remove the name here.

P3L1-10: It needs to be discussed here where the new shape parameter is more realistic than the previous one (based on literature), at which latitudes and at which temperature (ranges). In particular it needs to be shown that it's more realistic for the SO region.

P3L18: Do not use contractions in scientific writing.

P4L22: The shear-dominated unstable layer shows a similar change in LWP in the cap experiment as the stratocumulus boundary layer type clouds. This needs to be discussed.

P4L29-31: The change in the shape parameter will slow down the WBF process but the changes to nucleation temperature will determine if there is ice in the first place, which will also impact the WBF process. So why does the change in the shape parameter affect both LWP and IWP while changes to nucleation temperature mainly impact IWP?

P4L31: LWPs in Fig. 1 are for stratocumulus boundary layer type clouds, why are these dominated by fronts?

P5L26: Even in the c_tnuc=-40 simulation the SW TOA and surface biases compared to observations remain although there will be no more ice in mixed phase clouds. Therefore also other clouds should also contribute to these biases.

P5L32-33: Is it cloudy and non-cloudy grid boxes or cloudy and non-cloudy parts of grid boxes?

P6l7: control model, control run, ctrl experiment: choose one expression and use it consistently everywhere

P6L7-8: Not for the SO region (50° S to 70° S). What is the net bias over this region?

P6L8-9: It is stated that SW CRE is reduced over SO, but in the title or abstract the opposite is stated. For which experiments is there an improvement? This needs to be quantified (see general comment).

P6L9-10: This is in contradiction to the previous sentence.

P6L11: Do not use contractions in scientific writing.

P6L11-13: Why is the effect of the changed shape parameter in the tropics so much smaller than in the SO region?

P6L12-13: Don't you mean increase in SW CRE here? Fig. 2 suggests an increase in SW CRE in the sensitivity experiments.

P6L16-17: From Fig. 4b it is not clear that this is actually the case.

P6L18: Does the model have convective LWP? Rephrase otherwise

P6L18-19: Rephrase this sentence as it is unclear.

P7L1: Only impacts on mixed-phase clouds are discussed but what is the impact of the new shape parameter on cirrus clouds? This needs to be discussed as well.

P7L2-11: This belongs to the introduction.

P7L4: It is unclear what "It" refers to.

P7L18: "frontal steady states": I'm not familiar with this expression

P7L20-21: As there are undesirable biases in the NH (which need to be discussed more), what is the conclusion for the shape parameter? Should the new shape parameter be used in global climate models or not?

P7L29-32: Which assumptions about the INP concentration are made?

P8L9-10: Has the bias compared to observations in the SO region (50° S to 70° S) indeed been reduced? This needs to be quantified (see specific comments above and general comment).

P15: Fig. 2: The use of upward fluxes for TOA and downward fluxes for the surface should be specified in the title of each panel. For all figures that show fluxes it should be specified in the panel titles if upward or downward fluxes are shown. In the caption of Fig. 2 it is mentioned that upward fluxes are used for TOA and downward fluxes for the surface. This information needs to be added to other figures that show upward or downward fluxes.

---

## Author Response (AR2)

**Reply to the editor**

We would like to thank again both the anonymous reviewers and Editor for reviewing this work.

We have addressed all the technical corrections suggested by Reviewer2 in the modified version.

The main point raised by Reviewer1 is about the compensating error from the LW TOA fluxes. To this point we have now added the biases and rmse values for these TOA fluxes in the Supplementary material for both Southern Ocean and global mean with respect to observational data. As we have now added in the modified Discussion section, the improvement in SW TOA always leads to an increase in the LW component due to thick clouds which is unfortunately unavoidable.

We have also made it clear that even if the same value for capacitance could be applied globally, the only regions where it will make noticeable differences are that of the mixed-phase cloud regions (like SO).

Below are the specific points that we have addressed.

1. Title: Is the Southern Ocean cloud albedo bias reduced compared to observations? This needs to be quantified, see general comment.

We have modified the title to 'Improving the Southern Ocean cloud albedo biases in a general circulation model' to avoid any confusion. The SW CRE bias values have been presented in Table S1 of Supplementary material.

2. P1L1-10: The SW changes are accompanied by LW changes; this needs to be mentioned in the abstract and main text and also be discussed in the main text.

We have modified the Discussion section to stress on LW compensating effect. We have also added the bias values along with the RMSE values (both SO and global) with respect to observational data in tables S2 and S3 of Supplementary material.

3. P2L18: Unless this a named sea surface temperature climatology remove the name here.

Removed

4. P3L1-10: It needs to be discussed here where the new shape parameter is more realistic than the previous one (based on literature), at which latitudes and at which temperature (ranges). In particular it needs to be shown that it's more realistic for the SO region.

Included in the Discussion section

5. P3L18: Do not use contractions in scientific writing.

Corrected

6. P4L22: The shear-dominated unstable layer shows a similar change in LWP in the cap experiment as the stratocumulus boundary layer type clouds. This needs to be discussed.

Addressed in the Supplementary section where the particular plot is shown

7. P4L29-31: The change in the shape parameter will slow down the WBF process but the changes to nucleation temperature will determine if there is ice in the first place, which will also impact the WBF process. So why does the change in the shape parameter affect both LWP and IWP while changes to nucleation temperature mainly impact IWP?

Modified the results section to make it clear.

8. P4L31: LWPs in Fig. 1 are for stratocumulus boundary layer type clouds, why are these dominated by fronts?

Sentence rephrased

9. P5L32-33: Is it cloudy and non-cloudy grid boxes or cloudy and non-cloudy parts of grid boxes?

Rephrased the sentence

10. P6l7: control model, control run, ctrl experiment: choose one expression and use it consistently everywhere

Corrected

11. P6L7-8: Not for the SO region (50S to 70S). What is the net bias over this region?

Same as response 1.

12. P6L8-9: It is stated that SW CRE is reduced over SO, but in the title or abstract the opposite is stated. For which experiments is there an improvement? This needs to be quantified (see general comment).

Same as response 1.

13. P6L9-10: This is in contradiction to the previous sentence.

The SW CRE is calculated by taking the anomaly of TOA SW flux between the clear-sky and all-sky conditions. The SW CRE shows a reduction (as shown in Fig 5) compared to the control run. Whereas Figs. 2b and 3b are only showing the outgoing SW radiation at the TOA which is showing an increase compared to the control run. Basically showing that there is now more outgoing SW at TOA over SO in the experiments w.r.t control run. So an increase in outgoing SW TOA or a reduction in SW CRE over SO would be improvement.

14. P6L11: Do not use contractions in scientific writing

Corrected

15. P6L11-13: Why is the effect of the changed shape parameter in the tropics so much smaller than in the SO region?

Already included in the Results section.

16. P6L12-13: Don't you mean increase in SW CRE here? Fig. 2 suggests an increase in SW CRE in the sensitivity experiments.

Same as response 13.

17. P6L16-17: From Fig. 4b it is not clear that this is actually the case.

Same as response 13.

18. P6L18: Does the model have convective LWP? Rephrase otherwise

Removed the sentence

19. P7L4: It is unclear what "It" refers to.

Modified

20. P7L18: "frontal steady states": I'm not familiar with this expression

Repharsed the sentence.

21. P7L20-21: As there are undesirable biases in the NH (which need to be discussed more), what is the conclusion for the shape parameter? Should the new shape parameter be used in global climate models or not?

Same as response 4.

22. P7L29-32: Which assumptions about the INP concentration are made?

Rephrased the sentence.

23. P8L9-10: Has the bias compared to observations in the SO region (50S to 70S) indeed been reduced? This needs to be quantified (see specific comments above and general comment).

Same as response 1

24. P15: Fig. 2: The use of upward fluxes for TOA and downward fluxes for the surface should be specified in the title of each panel. For all figures that show fluxes it should be specified in the panel titles if upward or downward fluxes are shown. In the caption of Fig. 2 it is mentioned that upward fluxes are used for TOA and downward fluxes for the surface. This information needs to be added to other figures that show upward or downward fluxes.

Added

[revised manuscript text omitted]